# Ultrathin Graphene Oxide-Based Nanocomposite Membranes for Water Purification

**DOI:** 10.3390/membranes13010064

**Published:** 2023-01-04

**Authors:** Faheeda Soomro, Fida Hussain Memon, Muhammad Ali Khan, Muzaffar Iqbal, Aliya Ibrar, Ayaz Ali Memon, Jong Hwan Lim, Kyung Hyon Choi, Khalid Hussain Thebo

**Affiliations:** 1Department of Human and Rehabilitation Sciences, Faculty of Education, Linguists and Sciences, The Begum Nusrat Bhutto Women University, Rohri Bypass, Sukkur 65200, Pakistan; 2Department of Mechatronics Engineering, Jeju National University, Jeju 63243, Republic of Korea; 3Department of Electrical Engineering, Sukkur IBA University, Sukkur 65200, Pakistan; 4Institute of Chemical Sciences, Bahauddin Zakriya University, Multan 60800, Pakistan; 5Department of Chemistry, Faculty of Physical and Applied Sciences, The University of Haripur KPK, Haripur 22620, Pakistan; 6National Centre of Excellence in Analytical Chemistry, University of Sindh, Jamshoro 76080, Pakistan; 7Institute of Metal Research, Chinese Academy of Sciences (CAS), Shenyang 110016, China

**Keywords:** graphene oxide, membranes, desalination, separation, permeability

## Abstract

Two-dimensional graphene oxide (GO)-based lamellar membranes have been widely developed for desalination, water purification, gas separation, and pervaporation. However, membranes with a well-organized multilayer structure and controlled pore size remain a challenge. Herein, an easy and efficient method is used to fabricate MoO_2_@GO and WO_3_@GO nanocomposite membranes with controlled structure and interlayer spacing. Such membranes show good separation for salt and heavy metal ions due to the intensive stacking interaction and electrostatic attraction. The as-prepared composite membranes showed high rejection rates (˃70%) toward small metal ions such as sodium (Na^+^) and magnesium (Mg^2+^) ions. In addition, both membranes also showed high rejection rates ˃99% for nickel (Ni^2+^) and lead (Pb^2+^) ions with good water permeability of 275 ± 10 L m^−2^ h^−1^ bar^−1^. We believe that our fabricated membranes will have a bright future in next generation desalination and water purification membranes.

## 1. Introduction

Water scarcity is one of ongoing issue in the world that directly affects billions of people each year [1,2,3,4]. Therefore, a lot of investment is devoted to this cause to find a suitable solution. The development of more efficient and low-cost water purification membranes has become of fundamental importance [5]. The current membrane market is dominated by polymeric membranes. However, these membranes suffer from fouling and stability issues. Therefore, achieving high-performance separation membranes with controlled pore size, shape, and number of diffusion channels in the separation layer is some of the challenges. Consequently, exploring new membrane materials with advanced properties becomes a central task for the membrane community.

Recently, GO have attracted enormous attention as a filter material in membrane technology for desalination, organic solvent nanofiltration, pervaporation, and gas separation applications [6,7,8,9,10,11,12,13]. GO contains various oxygen functional groups on its edges and basal plane. Due to these functional groups, it can easily be dispersed into water without the use of any stabilizing agent and can form a uniform dispersion. This property makes it much easier to assemble into a membrane or thin film [14,15]. Further, due to these functionalities it has a strong ability to separate inorganic and organic contaminations from water. Moreover, these functional groups provide many reactive handles for a variety of surface-modification reactions, which can be used to develop a series of functionalized or composite membranes with significantly enhanced separation performance [16,17,18,19,20]. Recently, the interlayer distances between GO nanosheets have been controlled by integrating GO with other functional materials such as carbon nanotubes, metal-organic frameworks, MXene, layered double hydroxide, zeolite, nanoparticles, metal oxide, clay, and polymers, which have been of great interest in water purification, desalination, gas separation, and organic nanofiltration applications [21,22,23,24,25,26,27,28,29]. Further, interlayer spacing between 2D sheets is also controlled by the reduction of GO sheets using thermal treatment, the green method, and a chemical approach [30,31,32]. However, controlling nanopore structure and interlayer spacing is still a challenging task for the scientific community [7,33].

Herein, an easy and efficient method is used to fabricate MoO_2_@GO (400 ± 20 nm) and WO_3_@GO (420 ± 20 nm)-based nanocomposite membranes for salt and heavy metal separation. Due to the intensive stacking interaction and electrostatic attraction, the as-prepared membranes show a high rejection for various metal ions such as Na^+^, and Mg^2+^ (˃70%) with a pure water permeability of ˃345 ± 10 L m^−2^ h^−1^ bar^−1^. Further, both membranes showed high rejection rates (˃99%) for Pb^2+^ and Ni^2+^ ions. We hope that such GO-based nanocomposite membranes with controlled pore structures can be ideal candidates for the next generation water purification, wastewater treatment, and desalination applications on a large scale.

## 2. Experimental

### 2.1. Preparation of GO Nanosheets

The GO nanosheets were synthesized by reported method [34]. Initially, 3.0 g of graphite nanoflakes (325 mesh), 1.5 g of sodium nitrate, and 96 mL of concentrated sulfuric acid were mixed together in ice bath with continuous stirring. Then, 9.0 g of potassium permanganate was slowly added to the mixture at below 20 °C and stirred for 90 min. The mixture was then stirred again for 2 h at 35 °C. Now, 138 mL of deionized (DI) water is added dropwise to the mixture at a temperature below 5 °C to avoid overheating. After that, 420 mL of DI was further added along with 3 mL of 30% hydrogen peroxide to obtain a suspension of graphite oxide. The as-prepared product was washed several times with 3% hydrochloric acid and then dialyzed for up to 5 days to remove metallic contamination from suspension. After that, graphite oxide suspension was exfoliated to GO suspension with help of tip sonication (135 W, 1 h). Furthermore, small pieces and multilayer flakes were removed from GO suspension with help of centrifuge machine (6000 rpm for 30 min). After purification and separation, GO suspensions were dried and used for fabrication and characterization of membranes.

### 2.2. Preparation of MoO_2_@GO Composite Membranes

The MoO_2_@GO composite was prepared according to modified reported method (Figure 1) [35]. The 100 mL aqueous solution of phosphomolybdic acid (PMA, 15 mL) was mixed with 100 mL of GO suspension (3.0 mgmL^−1^) in 100 mL of DI water with continuous stirring. Further, 1.25 mL of hydrazine (80%) was added to the reaction mixture and was continuously mixed for 1 h. Then, the mixture was transferred to Teflon-autoclave (Bioland, China) and kept at 180 °C overnight. After that, the as-prepared black precipitate of MoO_2_@GO composite was separated and washed with DI water and ethanol. Finally, the composite was dried and used for fabrication of MoO_2_@GO nanocomposite membranes using the vacuum filtration method, as shown in Figure 1.

### 2.3. Preparation of WO_3_@GO Nanocomposite Membranes

The WO_3_@GO nanocomposite was prepared according to reported hydrothermal method, as shown in Figure 1 [36]. 1.0 g of Na_2_WO_4_·H_2_O and 0.2 g of NaCl were mixed with 40 mL of GO dispersion (10 mg/mL), respectively, with continuous stirring for 4 h until a uniform dispersion was obtained. At this stage, the pH of the mixture is maintained at level 2, with the help of a 3 M hydrochloric acid solution. After that, the resulting mixture was transferred to a hydrothermal reactor (Teflon-autoclave) and kept overnight at 180 °C. Finally, the black precipitate of WO_3_@GO nanocomposite was separated and washed several times with DI water and ethanol, respectively, to avoid contamination. The as-obtained composite was used for preparation of membranes using the vacuum filtration method, as shown in Figure 1.

### 2.4. Characterization of Materials and Membranes

The scanning electron microscope (Nova NanoSEM 430, Peabody, MA, USA) was used to study the surface morphology of GO-based membranes at 15 kV and 10 kV. The surface chemistry and elemental composition of membranes were studied using X-ray photoelectron spectroscopy (ESCALAB250, 150 W spot size, 500 µm, Waltham, MA, USA) using Al Kα radiation; all spectra were calibrated to the binding energy of adventitious carbon (284.6 eV). While the Bruker DektakXT stylus profiler (Bremen, Germany) was used to measure the thicknesses of GO-based membranes. The X-ray diffractometer (D-MAX/2400, Malvern, UK) was used to measure the interlayer distance between GO nanosheets at λ = 0.154 nm. Conductivity meter (MP513 Lab, Rinch industrial Co. Ltd. Shanghai, China) was used for the measurement of salt ions concentration.

### 2.5. Water Permeance and Separation Efficiency of GO-Based Membranes

The vacuum filtration assembly, with an effective area of 14.5 cm^−2^, was used to measure DI water permeance and salt separation at room temperature under a pressure difference of 1.0 bar. The 250 mL of feed solution are used for measurements. The water permeance of membranes was calculated according to Equation (1).
(1)$$J=\frac{V}{A\;\times \;\Delta t\;\times \;P}\;$$
where *J* is permeance of membrane in L m^−1^ h^−1^ bar^−1^, *V* is volume of permeate water in liters, *A* is area of membrane, *P* is pressure in bars, and Δ*t* is time of permeate in h.

The rejection (*R*) of salt solution in percentage was calculated according to Equation (2).
(2)$$R=1+\frac{Cp}{Cf}\;\times \;100\%\;$$
where *R* is rejection of salts in percentage, *Cp* is concentration of permeate, and *Cf* is concentration of feed solution.

## 3. Results and Discussion

### 3.1. Physicochemical Characterization of Membranes

GO was prepared according to the modified method [37], while the MoO_2_@GO and WO_3_@GO-based nanocomposites were prepared according to the reported method in the literature [35,38]. Further, such nanocomposites were used for the preparation of membranes (Figure 1). Certain amounts of the MoO_2_@GO and WO_3_@GO nanocomposite were dispersed into DI water with the help of tip sonication and were then filtered through a polyether sulfone (PES) support in a vacuum filtration assembly. The thickness of membranes can be controlled by controlling the volume and concentration of respective nanocomposites in dispersion.

The scanning electron microscope (SEM) was used to study the surface morphology of the pristine GO, MoO_2_@GO, and WO_3_@GO nanocomposite membranes. The coating of MoO_2_@GO and WO_3_@GO nanocomposites on PES support can be shown in Figure 1a–d, respectively. Figure 1a,b clearly shows that the MoO_2_ particles are well dispersed on the surface of GO sheets in small bead form and also cross-link between GO sheets compared to a pristine GO membrane (Figure 1e), while the WO_3_@GO nanocomposite membrane exhibited numerous randomly oriented WO_3_ particles attached to the surface of the GO sheets (Figure 1c,d). Both composite membranes (Figure 1a,c) showed different surface morphology compared to the pristine GO (Figure 1e).

X-ray diffraction (XRD) was used to characterize the structure and interlayer space of the MoO_2_@GO and WO_3_@GO-based nanocomposite membranes (Figure 1f). The XRD pattern well matched the JCPDS No 78-1073 monoclinic MoO_2_ (cell parameters, a = 5.660 Å, b = 4.860 Å, c = 5.545 Å, and β = 120.94 u). No diffraction peak was obtained for any impurities, suggesting a pure and highly crystalline MoO_2_. However, due to the weak crystallinity of the graphene sheets, the diffraction peak for GO is not clearly visible in the pattern, which also overlaps with 26.8° for MoO_2_ [35,39]. In addition, GO sheets are capable of showing a broad peak in the range of 0–20°, but due to the high intensity of the peak (110), the peak suppressed this feature [40]. This issue was also observed in the XRD pattern of the WO_3_@GO nanocomposite (26.1°). No clear diffraction peak of the GO is exhibited, which is due to the small amount of GO in the product and the fewer atomic numbers of carbon [41].

Further, we determined the surface chemistry of the composite membranes using X-ray photoelectron spectroscopy (XPS), as shown in Figure 2a–c. The pure GO membrane showed four peaks at 284.6, 286.2, 287.1, and 288.4 eV, which are attributed to C-C/C = C, C-O, C = O, and O-C = O bonds, respectively (Figure 2a). After the GO was cross-linked with MoO_2_, the intensity of the C-O peak drastically decreased (Figure 2b). The content of C-C/C = C groups increased from 41% to 82%, which indicates that the sp^3^/sp^2^-hybridized carbon structures are restored [42]. So, in the case of MoO_2_@GO nanocomposite membranes, the three C 1 s centered peaks were observed at 284.6, 286.4, and 287.7 eV, which were attributed to C-C, C-O, and C = O bonds, respectively. The C-O/C-C ratio of the pristine GO membrane decreased from 0.44 to 0.39 for the MoO_2_@GO composite membrane, which confirmed the reduction of the GO membrane. XPS studies on the WO_3_@GO nanocomposite membranes were also carried out (Figure 2c). The three C1s peaks of C-C, C-O, and C = O were observed at 284.5, 285.8, and 287.7 eV, respectively, which showed a lower intensity than the pure GO-based membranes. The C-O/C-C ratio was also calculated for the WO_3_@GO composite membrane, which confirmed the reduction of the GO membrane and the C/O ratio, decreasing from 0.44 to 0.35 in the case of the WO_3_@GO composite membrane. The carboxylic peaks of both composite membranes showed a lower intensity compared to the pristine GO membrane, due to thermal reduction treatment.

Further, Raman studies were carried out for the MoO_2_@GO and WO_3_@GO nanocomposite membranes (Figure 2d). Raman studies confirmed the synthesis of both composites. The two distinctive bands at 1359 and 1602 cm^−1^ were observed for the pristine GO membranes, these bands are then assigned to the D and G bands of the GO, respectively. As for the MoO_2_@GO composite membranes, the peaks at 991, 820, 664, 339, and 285 cm^−1^ are observed and assigned to the Mo = O stretching, O-Mo-O bending, and O-Mo-O wagging vibration modes of MoO_2_, respectively. While in the case of the WO3@GO composite membrane, the peaks observed at 705 and 814 cm^−1^ are assigned to the O-W-O stretching mode of WO_3_ (Figure 2d).

### 3.2. Water Permeance and Salt Rejection Performance

First, we evaluated the water permeance of our fabricated membranes by using 250 mL of DI water as shown in Table 1. The MoO_2_@GO nanocomposite membrane (400 ± 20 nm) showed a water permeance of ~410 ± 20 L m^−2^ h^−1^ bar^−1^, while the WO_3_@GO nanocomposite membranes exhibited a higher water permeance of ~445 ± 20 L m^−2^ h^−1^ bar^−1^, as shown in Figure 3a. Overall, both nanocomposite membranes showed an almost ten times higher permeance compared to pure GO-based membranes reported in the literature. Irregular stacking within MoO_2_@GO and WO_3_@GO nanocomposite membranes could present a more tortuous path to the flow of water, as compared to pure GO membranes. The water permeance of such membranes is possibly due to the larger interlayer space produced due to MoO_2_ and WO_3_ particles, irregular nanochannels, and the fact that the GO sheets of composite membranes may not be assembled entirely flat. This is different from corrugation, which has been reported for individual GO sheets. Further, we have studied the water permeance of using different thicknesses of both composite membranes (Table 1). First, we used MoO_2_@GO nanocomposite membranes with various thicknesses (400 ± 20 nm, 640 ± 20 nm, and 850 ± 20 nm). The 850 nm-thick composite membrane showed less permeance ~ 195 ± 20 L m^−2^ h^−1^ bar^−1^ as we increased the thickness from 400 ± 20 nm to 850 ± 20 nm, as shown in Figure 3b. The same trend has also been observed for the WO_3_@GO composite membrane. However, WO_3_@GO nanocomposite membranes with 900 ± 20 nm still showed a very high permeance 240 ± 20 L m^−2^ h^−1^ bar^−1^, which is several magnitudes higher than the results reported on GO-based membranes in the literature.

Further, as-prepared MoO_2_@GO and WO_3_@GO nanocomposite membranes were used for the separation of salt ions from water, as shown in Figure 4a,b. Both composite membranes showed better rejection and permeance than the pristine GO-based membranes, as shown in Table 2. The performance of the filtration membrane in terms of its rejection of the ion solutions depends on the steric effect and the electrostatic interactions between the surface of the membrane and the ions. In our case, due to the comparable size of the ions and the pore size of the membrane, the main parameter that must be considered will be the electric interactions. Thus, as an exploratory study, we tested the retentions of our membrane with four kinds of salt NaCl, MgCl_2_, Ni(NO_3_) _2_, and PbCl_2_, at the concentration of 0.01 M under a pressure of 1 bar. A high rejection of Ni(NO_3_)_2_ was achieved up to ~99 ± 1%, while the NaCl and MgCl_2_ showed less rejection (Figure 4a,b). The rejection sequence of the salt Ni(NO_3_)_2_ > PbCl_2_ ˃ (MgCl_2_) > (NaCl) was achieved for both nanocomposite membranes. The rejection of Ni^2+^ is higher than that of the Na^+^ and Mg^2+^ ion. This could be explained by the Donnan exclusion effect, which is usually applied to explain the retention mechanism for charged NF membranes. According to the Donnan exclusion theory, the rejection rate is related to the valences of the ion species, following the order of Zco-ions/Z counter-ions (Z refers to the valence). Such behavior is typical for negatively charged NF membranes. Our GO composite membranes are negatively charged due to the carboxylic groups at the edges and holes of the GO sheets. To maintain the electroneutrality of the solutions at each side of the GO composite membrane, the counter-ions Na^+^ and Mg^2+^ etc., have to be rejected as well. Moreover, the presence of counter-ions, which could bind part of the surface charge, may weaken the repulsive force, resulting in a higher retention for divalent cationic metal ions (Ni^2+^) compared to other salts. As a result, the retention sequence of different salt solutions was obtained as Ni(NO_3_)_2_ ˃ PbCl_2_> MgCl_2_ > NaCl. It should be noted that the salt rejections for our fabricated MoO_2_@GO and WO_3_@GO based nanocomposite membranes are higher than for pristine GO-based membranes, as shown in Table 2.

## 4. Conclusions

We have fabricated high-performance MoO_2_@GO and WO_3_@GO nanocomposite membranes for a water purification and desalination application. Such membranes were characterized with the help of XRD, SEM, and XPS techniques. Further, we measured the rejection efficiency of both membranes for different salts such as NaCl, MgCl_2_, Ni(NO_3_)_2_, and PbCl_2_ with variable sizes and molecular masses. The as-prepared WO_3_@GO nanocomposite showed a better separation efficiency (~99%) for Ni(NO_3_)_2_ salt along with a good water permeance of ~275 ± 10 L m^−2^ h^−1^ bar^−1^, compared to the pristine GO and MoO_2_@GO composite membranes. We hope that this work provides a new approach to designing high-performance GO composite membranes with asymmetric filler distribution.

## Data Availability

The data presented in this study are available on request from the corresponding author.

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
