# Peer review of "Ultrathin Graphene Oxide-Based Nanocomposite Membranes for Water Purification"

_membranes, 2023, doi:10.3390/membranes13010064_

Round 1

Reviewer 1 Report

The authors tackle a topic of great interest in the environment remediation field. The synthesis and  characterization of the own graphene oxide in a reproducible way is a challenge. The results regarding the use as water purification membranes are promising. However, I consider that some interpretation about the nature of the composite should be revised. 

I have the following observations:

1. Since the obtention of the composite implies temperature higher than room temperature, the state of the art in the introduction should include reported evidences about changes in the nature of GO towards thermal reduction of GO. Regarding this point, the control of the interlayer spacing when GO is reduced, joined with the change in the rugosity could contribute to the statement "However, control nanopore structure and interlayer spacing is still a challenging task for  scientific community."(page 2, line 53). Please include appropiate references.

2. Is it feasible to include Raman spectra of initial GO and  GO after the composite obtention to detect any change ID/IG?

3. Characterization: I suggest that a SEM image of pristine GO should be include to better understand changes in the morphology due to the temperature. Regarding GO XRD, from literature, it has been reported a peak around 10°C. Do the authors consider that when GO is under thermal treatment, a gradual reduction could occur? This phenomena could be observed through the XRD pattern, with a peak towards higher 2 theta due to the loss of functional groups. Please discuss XRD patterns to analyse a potential reduction (see peak 2 theta 23°) and add references.

4. XPS: Could you please calcule the C/O ratio in pristine in comparison with the GO in the composites?

5. In case the previous analysis leads to different conclusions regarding the presence of GO or rGO, I suggest to update the denomination of the composite.

Kind regards

Reviewer 2 Report

Review of Manuscript “Ultrathin graphene oxide based nanocomposite membranes for water purification” by Faheeda Soomro et al.

 The manuscript deals with the preparation of GO-based nanocomposite membranes for the removal of salt and heavy metal ions, a subject that has attracted a lot of interest in the last years. One of the main challenges using GO based membranes is the control of the pore size and the interlayer distance that can affect the ions removal efficiency.

In this work the authors report interesting results achieved by MoO2@GO and WO3@GO nanocomposites, showing high rejection for Na+, Mg2+, Ni2+, Pb2+. However, before publication, some issues have to be clarified and some errors should be corrected, as listed below:

1) The results obtained for the composite membranes have been compared with results achieved for GO membranes in other literature papers. Why haven’t the authors prepared/used their own GO membranes as comparison with their composite materials?

2) line 60: at the end of the Introduction, results on Pb2+ rejection should be cited together with Ni2+

3) line 114: area in “cm2” instead of “cm-1

4) the membrane thickness in fig.1d looks much thicker than the values reported in fig.3 and in table 1...

5) Line 145, description of fig.2: Figure captions are wrong. Please correct "Fig.1a" with "Fig.2..." and add for each XRD peak in the figure the corresponding crystal planes for MoO2 and WO3 structures.  Furthermore, “XRD spectra” should be corrected into “XRD patterns”

6) Line 162: “Fig.1b” should be “Fig.2 …” (reorder the captions or the figures positions)

7) Check all the discussion on the XPS analysis and indicate the right figures (a, b, c, d). Furthermore, please, discuss why the peaks related to Oxygen functional groups (especially carboxylic groups) show a lower intensity for the composites with respect to GO

8) Line 175: Why didn't the authors compare the results obtained for composites with the ones obtained on their GO membranes? A comparison with the same GO without MoO2 and WO3 would be more significant, since GO materials can show some differences depending on the preparation methods.

9) Line 179: Can you demonstrate that interlayer space in the composites is larger than the one in GO? In the same line: “MoS2” should be “MoO2

10) Line 197: “comparable” instead of “compared”?

11) Lines 208-210: it is written that “Our GO composite membranes are negatively charged due to the carboxylic groups at the edges and holes of the GO sheets”, but XPS spectra showed a very low concentration of COOH groups in the composites, especially in WO3@GO ... Please explain and discuss it.

12) Line 215-217: “the salts rejections for our fabricated composite membranes are higher than to the reported GO-based membranes reported in literature as shown in Table 2”. Please, add in the table some examples of the literature results as comparison.

13) Line 223, Conclusion Section: I don’t find in the paper how the interlayer distance has been controlled or measured … XRD analysis should provide information, but the analysis reported here does not show the distance between GO planes. Therefore, a comparison with the composite layers is not possible.

14) There are many typos and English errors, for example: line 42 “can easily dispersed” should be “can be easily dispersed”, or lines 44-45 “it has strongly ability” should be “it has strong ability”, and so on. Please check the whole paper.

Round 2

Reviewer 1 Report

The authors have correctly addressed the suggestions. I consider that the revised version is suitable for publication in this journal.

Reviewer 2 Report

The authors have answered all the comments, but there is still a problem with the answer to my comment 7 that has to be fixed.

In the discussion of XPS results the authors state: “The C/O ratio of pristine GO membrane decreased from 0.44 to 184 0.39 for MoO2@GO composite membrane, which confirmed the reduction of GO membrane” and “The C/O ratio also calculated for WO3@GO composite membrane, which confirmed the reduction of GO membrane and C/O ratio decreased from 0.44 to 0.35 in case of WO3@GO composite membrane.”

If it was so, this would not indicate a reduction of GO when MoO2 or WO3 are present… Probably the authors wrote "C/O ratio" but I guess they mean the "C-O/C-C ratio". Please, check it!

After this correction, in my opinion the paper can be published.
